# Additional Metabolic Effects of Bariatric Surgery in Patients with a Poor Mid-Term Weight Loss Response: A 5-Year Follow-Up Study

**DOI:** 10.3390/jcm9103193

**Published:** 2020-10-01

**Authors:** David Benaiges, Maria Bisbe, Juan Pedro-Botet, Aleix de Vargas-Machuca, Jose M. Ramon, Manuel Pera, Montserrat Villatoro, Laia Fontané, Helena Julià, Elisenda Climent, Olga Castañer, Juana A. Flores-Le Roux, Alberto Goday

**Affiliations:** 1Departament de Medicina, Universitat Autònoma de Barcelona, 08139 Barcelona, Spain; mbisbe01@gmail.com (M.B.); 86620@parcdesalutmar.cat (J.P.-B.); aleixdevargasmachuca@gmail.com (A.d.V.-M.); 62257@parcdesalutmar.cat (H.J.); 61063@parcdesalutmar.cat (E.C.); 94066@parcdesalutmar.cat (J.A.F.-L.R.); Agoday@parcdesalutmar.cat (A.G.); 2Cardiovascular Risk and Nutrition Research Group (CARIN-ULEC), IMIM-Hospital del Mar, Barcelona, Biomedical Research Park (Parc de Recerca Biomèdica de Barcelona-PRBB), 08003 Barcelona, Spain; ocastaner@imim.es; 3Department of Endocrinology and Nutrition, Hospital del Mar, 08003 Barcelona, Spain; 93954@parcdesalutmar.cat (M.V.); lfontane@parcdesalutmar.cat (L.F.); 4Consorci Sanitari de l’Alt Penedès i Garraf, 08720 Vilafranca del Pendès, Spain; 5Unit of Gastrointestinal Surgery, Hospital del Mar, Institut de Recerca IMIM-Hospital del Mar, 08003 Barcelona, Spain; 16350@parcdesalutmar.cat (J.M.R.); 95405@parcdesalutmar.cat (M.P.); 6Centro de Investigaciones Biomédicas en Red de Obesidad y Nutrición, CIBERobn, 28029 Madrid, Spain

**Keywords:** bariatric surgery, obesity, severe obesity, weight loss, weight regain, sleeve gastrectomy, gastric bypass

## Abstract

To ascertain the 5-year metabolic effects of bariatric surgery in poor weight loss (WL) responders and establish associated factors. Methods: Retrospective analysis of a non-randomised prospective cohort of bariatric surgery patients completing a 5-year follow-up. Mid-term poor WL was considered when 5-year excess weight loss was <50%. Results: Forty-three (20.3%) of the 212 included patients were mid-term poor WL responders. They showed an improvement in all metabolic markers at 2 years, except for total cholesterol. This improvement with respect to baseline was maintained at 5 years for plasma glucose, HbA1c, HOMA, HDL and diastolic blood pressure; however, LDL cholesterol, triglycerides and systolic blood pressure were similar to presurgical values. Comorbidity remission rates were comparable to those obtained in the good WL group except for hypercholesterolaemia (45.8% vs. poor WL, *p* = 0.005). On multivariate analysis, lower baseline HDL cholesterol levels, advanced age and lower preoperative weight loss were independently associated with poor mid-term WL. Conclusions: Although that 1 in 5 patients presented suboptimal WL 5 years after bariatric surgery, other important metabolic benefits were maintained.

## 1. Introduction

Undoubtedly, bariatric surgery (BS) has proved to be the most effective treatment for morbidly-obese patients when conventional therapy has failed. Its health benefits have been shown to extend beyond weight loss, with improvement in or even remission of obesity-related comorbidities such as type 2 diabetes, hypertension or dyslipidaemia [1,2].

Several studies have shown that post-surgical weight loss varies widely among patients and a notable percentage can be qualified as poor weight loss (WL) responders [3,4,5,6]. They are more frequently identified at mid-term follow-up taking into account that most patients experience a slight weight regain from the first year post-surgery [3,7,8]. Despite being a widely recognised group, no previous studies focused on the evolution of metabolic parameters and comorbidity remission rates 5 years after BS. To our knowledge, the Farias et al. study is the only work comparing good and poor WL responders. Those authors only reported results on the remission rates of obesity-related comorbidities with a shorter follow-up (2 years) [6].

We can hypothesise that patients with poor WL at 5 years will also benefit from BS considering two aspects: firstly, weight loss is not the only factor responsible for the improvement in comorbidities in BS; other phenomena, such as hormonal mechanisms, have been established [9,10,11,12]. Secondly, patients considered as poor WL responders have been defined by a percentage excess weight loss (%EWL) < 50%. However, despite not achieving an optimal WL at 5 years, patient weight remained lower than that at baseline. Thus, the primary aim of the present study was to determine the benefits of BS in poor WL responders at 5 years, compare them with good WL responders and identify the pre-surgical factors associated with a poor WL response to BS.

## 2. Experimental Section

### 2.1. Study Protocol

A retrospective analysis was conducted of a non-randomised prospective cohort study of severely-obese patients undergoing BS at the Hospital del Mar, Barcelona from January 2004 to December 2014. Patients were between 18 and 60 years of age and met the 1991 BS criteria of the National Institutes of Health [13]. Patients with a body mass index (BMI) of 35–39 kg/m^2^ and metabolic abnormalities as well as patients with BMI > 40 kg/m^2^ were included. The indication for the type of surgical procedure (laparoscopic sleeve gastrectomy [LSG] and laparoscopic Roux-en-Y gastric bypass [LRYGB]) was based on clinical criteria and the consensus of the BS Unit. In this respect, LSG was preferred in younger patients, in those with a BMI 35–40 kg/m^2^, as a first-step treatment in cases with BMI > 50 kg/m^2^ (although given the positive LSG outcomes none of these patients had to further undergo LRYGB), and when drug malabsorption was to be avoided. Patients who did not complete a minimum of 5 years of follow-up were excluded. In accordance with the study protocol, all patients were evaluated preoperatively and at 3, 6, 12, 18 and 24 months post-surgery and annually thereafter by the same surgeon, endocrinologist and nutritionist. Protocol visits included measurements of weight, waist, hip circumferences, blood pressure, assessment of comorbidity status and laboratory tests for glucose, insulin, glycated haemoglobin (HbA1c), total cholesterol, high-density lipoprotein (HDL) cholesterol and triglyceride levels. All subjects provided their informed consent for the procedure and the study. The Ethics Committee of Parc de Salut Mar (2017/7722) approved the protocol in accordance with the ethical guidelines of the 1975 Declaration of Helsinki.

### 2.2. Anthropometric and Biochemical Measurements

BMI was calculated as weight in kilograms divided by the square of height in metres and the %EWL was based on weight in excess of that corresponding to a BMI of 25 kg/m^2^ for each patient. Poor WL responders were defined as those whose EWL did not reach 50% during post-surgical follow-up [3]. Patients whose %EWL at 5 years was under 50% were considered poor mid-term WL responders. Glucose was determined by the oxidase method. HbA1c was quantified by chromatography (Biosystem, Barcelona, Spain). Insulin was measured by radioimmunoassay (Insulin kit, DPC, Los Angeles, CA, USA). Homeostasis model assessment for insulin-resistance (HOMA-IR) indices was estimated using the following formula [14]: HOMA-IR = insulin (U/mL) fasting glucose (mmol/L)/22.5. Total cholesterol and triglycerides were determined using enzymatic methods in a Cobas Mira automatic analyser (Baxter Diagnostics AG, Düdingen, Switzerland). HDL cholesterol was measured using separation by precipitation with phosphotungstic acid and magnesium chloride and low-density lipoprotein (LDL) cholesterol concentration was calculated with the Friedewald formula [15]. In accordance with data from our population, the cut-off level for the HOMA-IR index to define insulin resistance was ≥3.29 [16]. Diabetes was considered when at least one of the following criteria was met: plasma glucose determination > 126 mg/dL, HbA1c > 6.5% or current treatment with antidiabetic medication or insulin. By contrast, diabetes remission was considered when plasma glucose < 100 mg/dL, HbA1c < 6.0% and no anti-diabetic medication was needed. Hypercholesterolaemia was defined as total cholesterol > 200 mg/dL or use of statin treatment. Remission was considered when total cholesterol was < 200 mg/dL in the absence of statin treatment. Hypertriglyceridaemia implied triglyceride levels > 150 mg/dL or fibrate treatment. Remission was considered when triglyceridaemia <150 mg/dL and no fibrate treatment was needed. Hypertension was defined as systolic blood pressure > 140 mmHg and/or diastolic blood pressure > 90 mmHg or need for antihypertensive treatment. Remission was considered when systolic < 140 mmHg and diastolic blood pressure < 90 mmHg and no antihypertensive treatment was needed.

### 2.3. Surgical Techniques

The LRYGB technique consisted of a 150 cm antecolic Roux limb with a 25-mm circular pouch-jejunostomy and exclusion of 50 cm of the proximal jejunum. In LSG, a longitudinal resection of the stomach from the angle of His to approximately 5 cm proximal to the pylorus was made using a 36 French bougie inserted along the lesser curvature. The same team of surgeons performed all operations [12].

### 2.4. Study Power Calculation

Accepting an alpha risk of 0.05 and a beta risk of 0.2 in a two-sided test, 135 subjects were necessary in the first group and 33 in the second to recognize as statistically significant a difference greater than or equal to 20 mg/dL in 5 years total cholesterol change. The common standard deviation for total cholesterol change was assumed to be 37 [17].

### 2.5. Statistical Analysis

Statistical analysis was calculated with IBM SPSS Statistics for Windows, Version 25.0 (Armonk, NY, USA: IBM Corp.). Data were expressed as percentages and frequencies for categorical variables, as mean ± standard deviation for continuous variables with a normal distribution and as median with interquartile range for continuous variables with a non-normal distribution. Normality was evaluated using the Kolmogorov-Smirnov test. For skewed variables, a logarithmic transformation was used to achieve normality. Fisher exact test or ×2 test was applied to determine the association between qualitative variables and Student t test to compare mean and standard deviations of quantitative variables Two-factor mixed ANOVA models were used to evaluate the evolution of continuous variables in each group and analyse differences between groups at each time point from baseline. Logistic regression analysiswas applied to evaluate factors independently associated with poor mid-term WL. All variables associated on univariate analysis (*p* < 0.1) with poor weight loss responders were included in the regression model. A *p* value < 0.05 was considered statistically significant. 

## 3. Results

One hundred and forty-four (40.4%) of the 356 patients who underwent BS between January 2004 and December 2014 were excluded due to lack of follow-up beyond 5 years. Therefore, 212 patients completed 5 years of follow-up and were included in this study. Baseline characteristics of the included patients are shown in Table 1.

As shown in Figure 1, the maximum proportion of good WL was present at 18 months post-surgery with a progressive decline thereafter. At 5 years, 43 patients (20.3%) presented EWL <50% and were considered poor mid-term WL responders. Baseline characteristics of poor WL responders at 5 years were comparable to good WL responders except for age, HDL cholesterol and type of surgery (Table 1). The %EWL and BMI trajectories of the two groups of subjects are depicted in Figure 2 and Figure 3, respectively. Differences between groups, in %EWL and BMI, were already apparent 6 months after surgery (*p* < 0.001) and remained significant throughout follow-up.

Changes in metabolic markers after BS in good and poor WL responders are shown in Table 2. Good mid-term WL responders showed a significant decrease in systolic and diastolic blood pressure, plasma glucose, HbA1c, HOMA-IR, total cholesterol, LDL cholesterol, and triglycerides and an increase in HDL cholesterol 2 years after BS. Improvement in these metabolic parameters was maintained at 5 years. Poor mid-term WL responders also showed an improvement in all metabolic markers at 2 years, except for total cholesterol. This improvement with respect to baseline was maintained at 5 years for plasma glucose, HbA1c, HOMA, HDL cholesterol and diastolic blood pressure; LDL cholesterol, triglycerides, systolic blood pressure became similar to pre-surgical values. When the 2- and 5-year evolution of these metabolic parameters was compared between good and poor WL responders, significant differences were found regarding 2- and 5-year total cholesterol changes and 5-year triglyceride variation.

When the remission rates of obesity-associated comorbidities were analysed, no significant differences were found between good and poor WL responders at 2 and 5 years post-surgery except for hypercholesterolaemia remission (Table 3).

On the multivariate analysis, lower baseline HDL cholesterol levels, advanced age and lower preoperative weight loss were independently associated with poor mid-term WL (Table 4).

## 4. Discussion

Different weight loss patterns were observed following BS. Although most participants maintained their weight loss over time, 20.3% had an EWL < 50% at 5 years. Interestingly, the present study found that, despite remaining in obesity range, improvements in several metabolic markers persisted at 5 years after surgery. Moreover, remission rates of several comorbidities associated with obesity showed no significant differences between good and poor mid-term WL. Predictors associated with a poor WL pattern were also identified in this study.

Despite the lack of uniform criteria to determine unsuccessful outcomes after BS, poor WL has been extensively defined as EWL < 50%. Most studies had short-term follow-up periods (1–2 years) and yielded highly variable poor WL rates, varying from 15 to 35% [4,5]. A study by de Hollanda et al., which included the same surgical techniques as the present study and a similar follow-up period, revealed a 24.3% rate of non-responsive patients [3].

Both groups presented clearly different weight loss trajectories. Of note, good WL responders reached on average a %EWL of approximately 80% between 1 and 2 years of follow-up and were subsequently able to remain within overweight range (BMI 25–30 kg/m^2^) until 5 years of follow-up. By contrast, poor WL responders achieved a lower maximum weight loss of around 60% and also had a greater weight regain so that mean BMI in poor mid-term WL responders was in class I obesity range (BMI 30–35 kg/m^2^) from 6 months to 2 years and worsened to class II obesity range (BMI 35–40 kg/m^2^) from 3 to 5 years of follow-up.

As expected, good mid-term WL in our cohort showed an improvement in all metabolic parameters and similar results to other study cohorts in which no distinction regarding weight response was taken into account [7,18,19]. Interestingly, poor WL also showed an improvement in all metabolic parameters at 2 years, except for total cholesterol. These results can be explained by the fact that the maximum weight loss in this group of patients was observed around this follow-up timepoint. In contrast, at 5 years, two different patterns emerged with some of these parameters worsening to levels similar to those at baseline and others maintaining the improvement at 5 years.

The fact that glucose, HbA1c and HOMA-IR did not worsen with weight gain suggests that the improvement in these metabolic parameters after BS was partially independent of weight. In this respect, amelioration in insulin resistance and glucose metabolism occurred soon after the procedure, when significant weight loss had not yet been achieved, suggesting the involvement of gut hormonal mechanisms such as increased secretion of incretins that enhance insulin sensitivity [9,10,11,12]. The other parameter that remained greater at 5 years compared with baseline despite weight recovery was HDL cholesterol. In this case, it seems that a reduction in insulin resistance and systemic inflammation favoured the production of large HDL particles which in turn improves cholesterol efflux [20].

In contrast, other parameters such as triglycerides, LDL cholesterol or systolic blood pressure showed worsening between 2 and 5 years after surgery in poor WL responders in line with the weight increase. For triglyceridaemia, the close relationship between triglyceride decrease and weight reduction after surgery is well known [21]. As for LDL cholesterol, no significant differences were found between good and poor WL responders. However, while LDL cholesterol levels at 2 and 5 years post-surgery remained lower than those at baseline in good responders, LDL in poor responders rose to levels similar to those at baseline from 2 to 5 years. These findings can be explained by the fact that the poor responder group did not as frequently undergo the LRYGB technique and the malabsorptive component seems to be the most significant factor in LDL cholesterol reduction following BS [22]. Finally, differences in total cholesterol were justified by the changes in LDL and HDL levels. While total cholesterol dropped in good responders owing to a notable decrease in LDL, total cholesterol in poor responders increased due to a rise in HDL cholesterol.

With regard to the clinical evolution of systolic and diastolic blood pressure, no differences were found between study groups at either 2 or 5 years. Despite that, systolic blood pressure increased after surgery in poor WL responders from 2 to 5 years, showing no significant differences with baseline values at 5 years. These results are in line with epidemiological studies that had reported a close relationship between weight loss and blood pressure reduction [23,24]. In a previous study of 197 patients who underwent RYGB or SG, relapse of hypertension at 3 years was observed in >20% of patients who had achieved apparent hypertension remission at 12 months [24]. The only independent predictor of relapse was the greater use of preoperative antihypertensive medication. This suggested a waning effect of BS on blood pressure control over time, although further high-quality data regarding the mid- and long-term effects on hypertension remission are required. In this respect, in the present study, diastolic blood pressure remained lower than baseline throughout follow-up, suggesting that systolic blood pressure appears to be more sensitive to weight regain than diastolic blood pressure [23]. The worsening of systolic blood pressure from 2 to 5 years in poor WL responders would especially harm subjects over 50 years of age, since above this age systolic blood pressure would be a greater predictor of cardiovascular events and in subjects <50 years old it would be diastolic blood pressure [25].

When comorbidity remission rates were compared, results pointed in the same direction as those obtained on analysing the evolution of metabolic parameters. Hence, no significant differences were found between the remission rates of good and poor WL responders, except for hypercholesterolaemia. Of note, hypertriglyceridaemia showed a greater tendency towards poor WL (*p* = 0.069), which concurs with the differences observed in the change in triglyceride levels at 5 years. However, these results contrast with those reported by Farias et al. [6] who compared good and poor responders after a 2-year period post-LRYGB. Significant differences in remission rates of type 2 diabetes, hypertension and hyperlipidaemia were found between good and poor responders. The distinct findings of both studies can be attributed to a shorter follow-up period, differences in sample size, LRYGB as the only surgical technique studied and a different proportion of poor responders.

Various predictors of weight change after BS were suggested in previous studies with variable magnitudes of effects [26,27,28]. A negative correlation has been reported between EWL and age, tobacco consumption, pre-operative BMI, female sex, diabetes and LSG, among others. In the present study, the only pre-surgical parameters showing a significant association were age, pre-surgical weight loss and baseline HDL cholesterol. These findings could be explained by the fact that ageing is associated with a progressive decline in the quantity and quality of muscle tissue, the basal metabolic rate and energy requirements. These metabolic changes could influence the response to BS [26]. Regarding pre-surgical weight loss, it seems reasonable to assume that patients who manage to lose more weight before surgery will be more efficient in maintaining weight loss over time. Other studies also favour this association [29,30]. On the other hand, HDL cholesterol has not been previously identified as a predictive factor of WL. However, some studies demonstrated a significant association between both moderate and vigorous physical activity and higher HDL concentration [31]. Hence, HDL could be interpreted as an exercise marker, thereby indicating that subjects with higher HDL concentrations and thus physically more active had a clear trend towards greater weight loss patterns. The differences between this study and others in identifying predictive factors could partially be explained by differences in baseline characteristics or how weight loss was defined (as a continuous variable or used as a cut-off point to define poor WL).

The present study had some limitations. There was a lack of follow-up of around 40%, which may have influenced the final results since the sample of patients used could not have been representative of the initial patient cohort. Nevertheless, the loss of a follow-up rate was similar to that of other studies with a 5-year follow-up conducted in other countries with a National Health Service such as ours. Unfortunately, no other factors that may have contributed to a poor WL, such as physical activity, eating behaviour, quality of life or interpersonal support were taken into account. Moreover, other comorbidities such as obstructive sleep apnea were not studied.

## 5. Conclusions

One out of 5 patients undergoing BS will fail to achieve and maintain an optimal WL 5 years after surgery. Despite what may be expected, these patients obtain and preserve clinical benefits, such as improvement in glucose and lipid metabolism, comparable to those achieved by good WL. In view of these data, it could reasonably be argued that in some patients BS, despite achieving slight weight loss, continues to provide other major metabolic benefits. These findings support the idea of changing the concept of bariatric surgery to metabolic surgery.

## Figures and Tables

**Figure 1 jcm-09-03193-f001:**
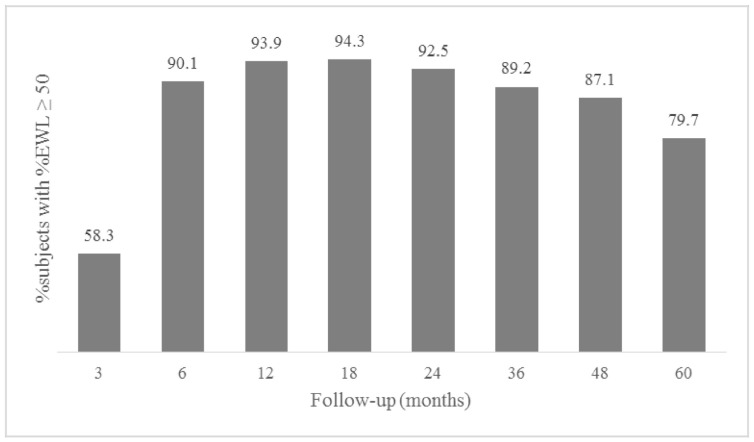
Percentage of good weight loss responders throughout follow-up. %EWL = percentage excess weight loss.

**Figure 2 jcm-09-03193-f002:**
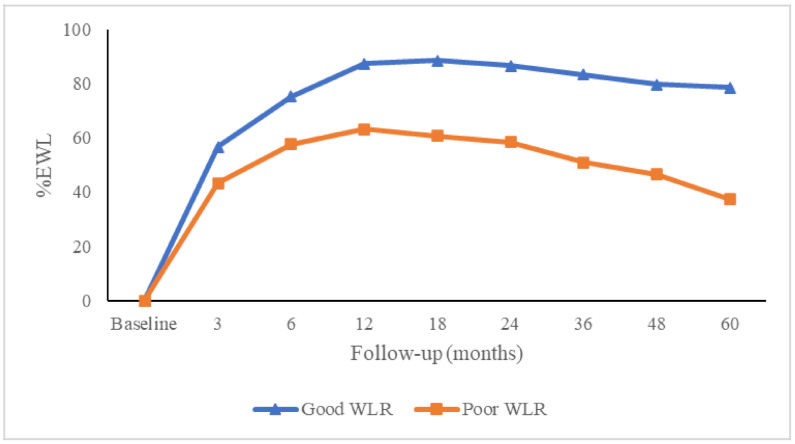
Changes in percentage of excess weight loss in good and poor weight loss responders during follow-up after the bariatric surgical procedure. %EWL = percentage excess weight loss; WLR = weight loss responders.

**Figure 3 jcm-09-03193-f003:**
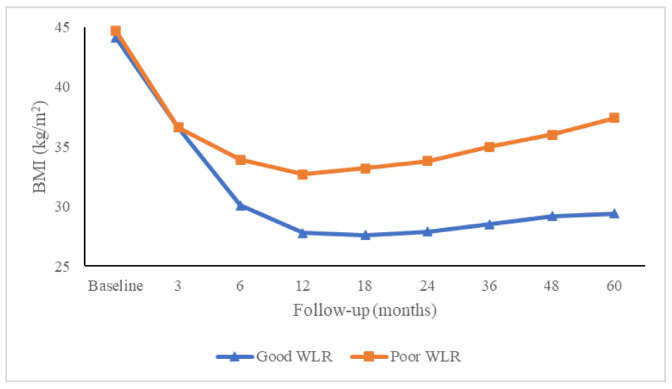
Changes in BMI of good and poor weight loss responders during follow-up after the bariatric surgical procedure. BMI = body mass index; WLR = weight loss responders.

**Table 1 jcm-09-03193-t001:** Baseline characteristics of all included patients, good and poor mid-term weight loss responders.

	Included Patients (*n* = 212)	Good Mid-Term WL Responders (*n* = 169)	Poor Mid-Term WL Responders (*n* = 43)	*p* Value *
Age (years)	45.4 ± 8.9	44.7 ± 8.9	48.4 ± 8.4	0.015
Female (%)	176 (83.0)	140 (82.8)	36 (83.7)	0.548
BMI (kg/m^2^)	44.2 ± 5.0	44.1 ± 4.9	44.7 ± 5.5	0.482
LRYGB (%)	131 (61.8)	110 (65.1)	21 (48.8)	0.038
Preoperative %EWL (%)	12.1 ± 13.0	12.9 ± 12.7	8.7 ± 13.5	0.064
Abdominal waist (cm)	126.3 ± 11.9	125.9 ± 12	127.7 ± 11.6	0.468
Systolic BP (mmHg)	139 ± 19.5	138.6 ± 20.0	140.6 ± 17.8	0.538
Diastolic BP (mmHg)	85.9 ± 11.7	86.12 ± 11.9	85.1 ± 11.0	0.603
Glycaemia (mg/dL)	118.4 ± 40.4	116 ± 41.6	127.7 ± 33.9	0.089
Insulin (mU/mL)	16.8 ± 14.4	16.8 ± 15.5	16.9 ± 9.0	0.959
HOMA-IR	5.5 ± 7.7	5.5 ± 8.4	5.5 ± 3.5	0.999
HbA1c (%)	5.97 ± 3.42	5.98 ± 3.8	5.93 ± 0.79	0.944
Total cholesterol (mg/dL)	194.8 ± 34.1	195.5 ± 34.5	191.72 ± 32.7	0.512
HDL cholesterol (mg/dL)	50.7 ± 15.6	51.9 ± 16.3	45.9 ± 11.5	0.023
LDL cholesterol (mg/dL)	118.6 ± 33.0	118.4 ± 34.2	119.6 ± 28.4	0.828
Triglycerides (mg/dL)	106.5 (82.3–151.5)	119.0 (90.0–146.0)	104.0 (77.5–152.5)	0.583
Hypertension * (%)	93 (43.9)	77 (45.6)	16 (37.2)	0.209
Anti-hypertensive drugs	77 (36.3)	62 (36.7)	15 (34.9)	0.826
Diabetes (%)	53 (25)	38 (22.5)	15 (34.9)	0.072
Oral antidiabetic drugs	26 (12.3)	15 (8.9)	11 (25.6)	0.002
Insulin	5 (2.4)	5 (3.0)	0	0.253
Dyslipidaemia (%)	61 (28.8)	45 (26.6)	16 (37.2)	0.120
Statins	31 (14.6)	21 (12.4)	10 (23.3)	0.065
Fibrates	12 (5.7)	9 (5.3)	3 (7.0)	0.454
Cigarette smoking (%)	58 (27.5)	47 (28.0)	11 (25.6)	0.458

WL = weight loss; BMI = body mass index; LRYGB = laparoscopic Roux en Y gastric bypass; %EWL = percentage excess weight loss; BP = blood pressure; HbA1c = glycated haemoglobin; HDL = high-density lipoprotein; HOMA-IR = homeostasis model assessment for insulin resistance; LDL = low-density lipoprotein. * *p* values for comparisons between poor and good weight loss.

**Table 2 jcm-09-03193-t002:** Evolution of metabolic markers of good and poor WL at baseline, 2 and 5 years after bariatric surgery.

	Good Mid-Term WLR (*n* = 169)					Poor Mid-Term WLR (*n* = 43)					Good vs. Poor WLR	
											2 Years	5 Years
	Baseline	2 Years	* *p* Value	5 Years	† *p* Value	Baseline	2 Years	* *p* Value	5 Years	† *p* Value	*p* Value	*p* Value
Glucose (mg/dL)	116.8 ± 43.1	90.0 ± 11.5	<0.001	91.7 ± 11.1	<0.001	126.4 ± 33.5	93.8 ± 12.1	<0.001	98.4 ± 13.1	<0.001	0.244	0.379
HbA1c (%)	5.8 ± 1.1	5.3 ± 0.6	<0.001	5.5 ± 0.6	<0.001	6.0 ± 0.8	5.4 ± 0.4	<0.001	5.5 ± 0.4	0.018	0.496	0.261
HOMA	5.8 ± 10.0	1.0 ± 0.8	<0.001	1.1 ± 0.9	<0.001	5.6 ± 3.4	1.7 ± 1.7	<0.001	2.3 ± 1.9	<0.001	0.701	0.447
Cholesterol (mg/dL)	197.4 ± 34.1	182.2 ± 31.8	<0.001	190.4 ± 34.9	0.015	192.3 ± 33.5	187.5 ± 34.7	0.714	196.9 ± 35.7	0.107	0.018	0.027
HDL cholesterol (mg/dL)	52.5 ± 17.1	68.7 ± 16.4	<0.001	70.7 ± 17.4	<0.001	45.5 ± 11.6	61.7 ± 14.3	<0.001	60.7 ± 18.6	<0.001	0.926	0.382
LDL cholesterol (mg/dL)	118.8 ± 35.5	98.8 ± 28.9	<0.001	103.5 ± 29.1	<0.001	120.8 ± 28.2	109.9 ± 27.1	0.016	114.3 ± 30.5	0.114	0.073	0.130
Triglycerides (mg/dL)	104.0 (77.5–52.5)	66.0 (53.0–83.0)	<0.001	70.0 (58.0–96.5)	<0.001	119.0 (90.0–146.0)	84.0 (69.0–115.0)	<0.001	90.0 (73.0–140.0)	0.243	0.109	0.006
Systolic BP (mmHg)	141.2 ± 21.0	120.4 ± 17.6	<0.001	128.4 ± 17.4	<0.001	143.8 ± 16.9	122.3 ± 15.5	<0.001	134.4 ± 16.8	0.072	0.340	0.131
Diastolic BP (mmHg)	86.6 ± 12.5	72.5 ± 11.4	<0.001	75.4 ± 10.6	<0.001	87.2 ± 11.4	75.3 ± 8.6	<0.001	78.6 ± 11.5	0.001	0.074	0.085

WLR= weight loss responders; BP = blood pressure; HbA1c = glycated haemoglobin; HDL= high-density lipoprotein; HOMA-IR = homeostasis model assessment for insulin resistance; LDL = low-density lipoprotein. * *p* values for comparisons within study group between baseline and 2 years. † *p* values for comparisons within study group between baseline and 5 years.

**Table 3 jcm-09-03193-t003:** Remission rate of metabolic comorbidities in good and poor weight loss responders at 2 and 5 years post-surgery.

	Good WLR at 2 Years	Poor WLR at 2 Years	* *p* Value	Good WLR at 5 Years	Poor WLR at 5 Years	† *p* Value
Type 2 diabetes (%)	32/38 (84.2)	14/15 (93.3)	0.351	27/38 (71.1)	11/15 (73.3)	0.577
Hypertension (%)	40/77 (51.9)	7/16 (43.8)	0.374	37/77 (48.1)	7/16 (43.8)	0.486
Hypercholesterolaemia (%)	44/83 (53.0)	6/22 (27.3)	0.027	38/83 (45.8)	3/22 (13.6)	0.005
Hypertriglyceridaemia (%)	21/25 (84.0)	4/6 (66.7)	0.327	22/25 (88.0)	3/6 (50.0)	0.069

* *p* value for comparison between study groups at 2 years. † *p* value for comparison between study groups at 5 years.

**Table 4 jcm-09-03193-t004:** Multivariate analysis of factors associated with poor mid-term WL after bariatric surgery.

	Odds Ratio	95% CI	*p* Value
Age (for each 5 years)	1.429	1.100–1.857	0.007
Baseline HDL cholesterol (for each 5 mg/dL)	0.753	0.629–0.901	0.002
Type 2 diabetes mellitus	1.009	0.445–2.287	0.983
Sleeve gastrectomy	1.643	0.798–3.383	0.178
Preoperative %EWL (for each 5%)	0.832	0.720–0.963	0.014

CI = confidence interval; HDL = high-density lipoproteins; %EWL = percentage excess weight loss.

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
