# Peer review of "Additional Metabolic Effects of Bariatric Surgery in Patients with a Poor Mid-Term Weight Loss Response: A 5-Year Follow-Up Study"

_jcm, 2020, doi:10.3390/jcm9103193_

Round 1

Reviewer 1 Report

Dear authors,

Thank you for the opportunity of reviewing your very interesting manuscript. I have a few comments:

1- Understanding that analysis of subgroups may decrease the opportunity of finding significant differences, I'd like to know if you run any analysis comparing subgroups such as patients with BMI lower than 40 vs patients with higher than 40. Maybe it's an opportunity to detect more differences the higher the initial BMI despite of "unsuccessful" metabolic surgery.

2- The trend is to change the concept of bariatric surgery completely to metabolic surgery, again, this is supported by your manuscript findings, highlighting the impact of surgery not only on weight but also on comorbidities improvement that is very well known.

3- I'd like to know if you find differences when Sleeve patients were compared to Gastric Bypass patients. It is known that comorbidities resolution is improved with bypass. Would be interesting to see that this occurs even when surgery failed to achieve goals of weight loss. Again, risk of comparing 2 smaller populations would miss differences. 

4-Regarding the impact of blood pressure and the differences in systolic and diastolic values would be interesting for the reader to have a reference related to which one has more impact in long term risk of cardiovascular disease. 

Reviewer 2 Report

Bariatric surgery has been proven to be the most effective treatment against obesity. Patients undergoing BS present a considerable weight loss and an amelioration in most of the metabolic parameters. However, an important number of BS patients don’t show that expected WL. These are considered as “poor responders”. Despite the substantial percentage of poor WL responders, not many studies have focused in the analysis of these individuals.

The present manuscript provides a results that will help to better understand the causes of the poor reponse to BS. Their metabolic analysis in more than 200 patients strongly shows a solid correlation between “poor WL responder” and cholesterol levels and other few parameters.

Overall, the study is well designed, the number of patients studied make the study very solid, and the results of metabolic analysis are clear and well explained.

Figure legends should be presented below the figures and above the tables. it causes a bit of confusion when reading it

Author Response

We are very grateful with the Reviewer’s positive evaluation. Figure legends are presented following JCM instructions for authors.

Reviewer 3 Report

This is an extremely nice cohort that answers a relevant question, i.e. does less weight loss impede on the maintanance of metabolic parameters. Overall I would like to see or at least discuss with the authors following issues

Major

  1. Although EWL is often used as a parameter in surgical literature, I find it less inuitive than absolute weight loss or drop in BMI, in addition the < 50% EWL cut-off is not extensively justified. Could the authors comment on this, including on the rationale for this specific cut-off
  2. In essence the authors claim that metabolic improvement in patients experience less weight loss (poor weight loss responders) is non-inferior in terms of metabolic outcome to those who experience good weight loss. What I am missing a bit is a power calculation, what is the power
  3. What happens with sleep apnea and is there a difference between groups

Author Response

This is an extremely nice cohort that answers a relevant question, i.e. does less weight loss impede on the maintanance of metabolic parameters. Overall I would like to see or at least discuss with the authors following issues

Major

Although EWL is often used as a parameter in surgical literature, I find it less inuitive than absolute weight loss or drop in BMI, in addition the < 50% EWL cut-off is not extensively justified. Could the authors comment on this, including on the rationale for this specific cut-off

We agree with the Reviewer that there is no consensus on which weight loss parameter and which cut-off point are best suited to define which subjects are PWL. As already reflected in the first version of the manuscript (lines 199-203), the fact of choosing <50% EWL cut-off is justified mainly because it is the one that has been used previously in more studies and this allows comparisons between them. In our case, the use of this cut-off point is reinforced when we verify that the subjects with an EWL> 50% had a mean BMI <30Kg/m2 during follow-up and therefore below the obesity range.

In essence the authors claim that metabolic improvement in patients experience less weight loss (poor weight loss responders) is non-inferior in terms of metabolic outcome to those who experience good weight loss. What I am missing a bit is a power calculation, what is the power

We have added a statistical power calculation to be able to detect differences in total cholesterol change between the groups (see revised manuscript, lines 113-117).

What happens with sleep apnea and is there a difference between groups

We believe that the Reviewer's comment is very appropriate since the resolution of sleep apnea is clearly related to weight loss. Unfortunately this variable was not collected in our study. We have added this as a limitation (see revised manuscript, line 290).